# A Personal View of Microstructure and Properties of Al Alloys

**DOI:** 10.3390/ma14051297

**Published:** 2021-03-08

**Authors:** John Campbell

**Affiliations:** School of Metallurgy and Materials, Engineering Faculty, University of Birmingham, Birmingham B15 2TT, UK; jc@campbelltech.co.uk

**Keywords:** Al alloy, bifilm, properties, cracks, Si modification, textures

## Abstract

This paper presents a personal view by the author of the role of bifilms in Al alloys. The mantra ‘microstructure determines properties’ is widely accepted as a truism, but is here critically assessed and found wanting. The case is made that bifilms from the casting process, while often invisible in the microstructure, are usually at least as important, if not of far greater importance, because they are often present as a dense population of cracks throughout the metal. The bifilm population controls the morphology of many features of cast and wrought structures. For cast alloys, bifilm control of pore morphology and Si morphology in Al–Si alloys is discussed, as is dendrite arm spacing (DAS). The tensile property benefits of grain refinement are seen to be mainly bifilm controlled. The properties ductility and fatigue appear to be especially dominated by bifilm content, as are invasive corrosion processes such as pitting, intergranular corrosion, hydrogen blistering and cracking. Bifilm control is proposed as a new concept permitting the improvement and control of metallurgical properties.

## 1. Introduction

The metallurgical community has been largely unaware of the extent to which the casting process can affect the metal as an engineered product. The ignorance is a legacy of our preoccupation in our educational institutions with physical metallurgy and its undoubted successes, and the paucity of process metallurgy currently taught. This short article has much, therefore, to cover. For detail, therefore, the reader is recommended to consult the references.

The central issue is the consolidation concept [1], in which metals are consolidated from powders or larger fragments, or by from the splashes and droplets of liquid metals when poured to make a casting. In all of these ‘bringing together’ events to create pieces of metal sufficiently large to be useful, each contributing fragment is usually covered with an oxide film. The impact of particles or drops of metal, therefore, necessarily occurs oxide-to-oxide. Furthermore, it is the dry outer surfaces of the oxide which meet, so bonding cannot be expected between the two dry, ceramic surfaces. Conversely, the double oxide layer which is created, which I call a ‘bifilm’, has outer surfaces which are in atomic contact with the metal, i.e., are perfectly ‘wetted’ because these surfaces represent those interfaces from which the oxide layer grew atom by atom. The bifilm structure is, therefore, unique, exhibiting practically zero bonding between its interior faces, but perfect bonding of its exterior faces with its matrix. 

During powder metallurgy processing, therefore, each powder particle creates a bifilm between itself and its neighbour, creating a dense population of bifilms of similar size to the originating powder. The population is, however, extremely uniform and, therefore, has predictable properties.

When pouring liquid metals, the drops and splashes are of varying size, so that bifilms of widely variable size can be produced, explaining the statistical scatter for which castings are regrettably notorious. Such bifilms are common features in cast houses, explaining the massive cracks seen from time to time across the width of cast Al alloy slabs a meter wide, the subsequent explosive cracking of cast high strength logs, or their fracture during forging, or edge cracking during rolling, or crocodile cracking in extrusion. Bifilms significantly impair our metallurgical processing. The beautiful shining sheets and extrusions which finally get to market are often achieved at a high cost of failure; the industry operates with high values of internally generated scrap. 

In the liquid state, although aluminium oxide is denser than the liquid metal, and should cause bifilms to sink, the small layer of entrapped air and bubbles in the ‘air gap’ between the films causes them to have nearly neutral buoyancy. They, therefore, remain in suspension for hours or days, reducing properties. (This behaviour contrasts with dense alloys such as steels, in which bifilms float out quickly, conferring over ten times more elongation to failure, among other benefits to steels.) 

It seems worthwhile, therefore, to examine the problems from their source. The development of properties can then be interpreted more clearly.

In this personal overview of the already broad subject of microstructure and properties, the role of bifilms permeates nearly every aspect, permitting important new insights into the mechanisms underlying much of our metallurgy. Overviewing the sections listed here, Section 2 explains the necessity for bifilms to initiate porosity; Section 3 explains how our views of the effect of dendrite arm spacing (DAS) have been misled, with further confusion explained in Section 4 of the mechanism of improvement when grain refining.Section 5 discusses how other features of grains and their boundaries relate to bifilms, leading on to the long Section 6 on the structure of Al–Si alloys. This is a complicated section, but its complexity is now understandable for the first time. Other microstructural features such as texture (Section 7), and properties such as tensile elongation (Section 8) and fatigue (Section 9) are explained and linked. Even the action of hydrogen (Section 10) to embrittle or blister becomes clear. We are witnessing a new interpretation of our microstructures and properties of Al alloys.

## 2. Porosity

Porosity is an unwelcome component of many cast structures. This is especially true in shaped casting where the problem is often labelled ‘gas’, and is dealt with by some kind of hydrogen degassing process. However, the ‘gas’ is almost always air bubbles, entrained into the liquid metal during pouring [2]. The bubbles are not round as a result of them experiencing a punishing level of turbulence, and possibly contributed by the shrinking of the bubble by loss of oxygen by the thickening of the bubble’s oxide skin, which might later be further thickened by aluminium nitride after the oxygen is gone [3].

There is, however, genuine hydrogen porosity in cast Al alloys, which usually survives to some degree in wrought products. Nevertheless, this ‘true’ hydrogen porosity is generally at least an order of magnitude finer than the entrained air bubbles, and necessarily is always associated with bifilms. The bifilm avoids the difficulty of nucleation; the gas in solution diffuses to and precipitates into the ‘air gap’ of the bifilm, which easily separates its nearly parallel surfaces, and may inflate, tending to become spherical. It seems all gas porosity from hydrogen in solution necessarily forms on bifilms [4,5].

The spherical pores are often interpreted as ‘hydrogen gas pores’ while pores deformed by tips of dendrites are interpreted as ‘shrinkage pores’. This is not correct. The rounded pores are simply those that have formed early during freezing, and so have grown freely in the liquid. The interdendritic pores are those which have formed late, after the formation of dendrites. Both types of pores could represent bifilms which have been inflated by either hydrogen or shrinkage. The inflation of bifilms can be easy or difficult depending on how the bifilms are randomly scrambled during turbulence. Thus, a mix of rounded and dendritic pores are often found together (Figure 1) which would be difficult to explain by traditional nucleation and growth mechanisms [2]. 

The occurrence of so-called ‘secondary porosity’ in wrought Al alloys subject to extended times at high temperatures is reviewed by Talbot [6]. He presents much evidence how these micrometer-sized pores are concentrated in grain boundaries, although some of his evidence suggests *not all* grain boundaries. It may be that pores only precipitate on bifilms (not on grain boundaries as is commonly supposed) because of the ease of the opening of the bifilm, effectively avoiding the high strain energy (to deform the surrounding solid) required for nucleation and growth of a new precipitate.

## 3. Dendrite Arm Spacing

In common with general usage, although strictly this section should use the full description ‘secondary dendrite arm spacing (SDAS)’ the simpler DAS will be used as a convenient shorthand. 

The general relationship in which finer DAS gives higher tensile properties, as typified by the famous result by Miguelucci [7] shown in Figure 2, is well known, and widely used in the form of a specification to meet a required specific fineness of DAS to indicate the casting has adequate tensile properties. 

The link between DAS and properties appears to be generally assumed to be linked to the Hall–Petch relation, in which the length of free dislocation glide is limited by mechanical boundaries, such as a grain boundary, or in some cases, an interdendritic barrier of some kind such as Si particles in an Al–Si alloy. The limited length of slip is related to yield strength. However, as is clear from Figure 2, Miguelucci’s results show that yield strength is hardly affected by DAS. This is to be expected because the barriers between dendrites in the Al-7Si alloys are far from complete. Furthermore, when considering solid-solution alloys, there are zero barriers between dendrite arms, so that the Hall–Petch relation cannot apply at all. If this level of disagreement were not bad enough, elongation and UTS are much more importantly affected, which cannot be explained by the Hall–Petch relation. This all represents a serious logical oversight in traditional metallurgical thinking. 

A simple and elegant explanation can be found taking account of the presence of a population of bifilms in the metal. When first poured, the turbulence of pouring causes the bifilms to be compacted in untidily ravelled masses. These small features have little effect on properties, so properties are fairly good. If the metal is chilled rapidly, this structure can be frozen in place. Such rapid solidification will also result in fine DAS. For longer freezing times, however, the bifilms progressively unravel, eventually becoming large planar features resembling engineering cracks. Properties have been progressively reduced during this steady transformation. At the same time, of course, the DAS has been steadily growing because of the coarsening reaction to reduce their total surface energy. Thus, bifilms slowly straighten to reduce properties while DAS coarsens to reduce energy. The two are not causally related. The DAS is the independent clock which monitors the progress of the straightening bifilms. 

There is direct evidence for this explanation in the work of Polich and Flemings [8], as explained in [2]. In their research into a comparison of equiaxed and directionally solidified steels, elongation naturally falls in normal polycrystalline metals as DAS increases. However, in directionally solidified metal, in which the pushing action of the dendrites reduces the bifilm concentration, they find that ductility remains high and constant as DAS increases. 

In the future, when bifilm contents in our metals may be far better controlled, the consequent high properties would remain unaffected by the solidification time. The use of DAS as a quality control parameter would become redundant.

## 4. Grain Refinement

The mechanical properties of many cast Al alloys appear to be enhanced by the addition of grain refiners which nucleate new grains, and at the same time may also act to reduce the rate of growth of grains. However, because of the numerous slip planes available to the perfectly face-centred-cubic (FCC) lattice of aluminium, any benefit from the Hall–Petch relation is expected to be relatively poor, being nearly negligible in most alloys. Yet, in opposition to the prediction of the Hall–Petch relation, good benefits to tensile properties are widely observed. 

This seems to be the consequence of the relatively heavy Ti compounds precipitating on bifilms as favoured substrates, and thereby encouraging bifilms to sink. If the cleaned metal is then poured carefully (avoiding disturbance to the layer of bifilm and Ti-rich sediment) then superior properties are achieved principally as a result of the cleaning action—although the Hall–Petch relation would probably contribute a smaller proportion of the benefit [2].

The fact that in addition to the benefits to yield strength, the tensile elongation to failure is also significantly raised, proves that the Hall–Petch relation is not the dominant factor in the property changes because the Hall–Petch relation cannot explain the increased tensile elongation. This unexpected but widely overlooked result points to the presence of an additional important factor. It is the result to be expected from a reduction in bifilms, consequent on their sedimentation by grain refiners. This is now standard procedure in a growing number of Al alloy foundries as part of the system to achieve clean, high-property Al alloy products [2].

## 5. Grains and Grain Boundaries and Facets

In the equiaxed grain structures mentioned above, this most common mode of freezing traps bifilms between the growing grains because the bifilms are pushed ahead of the advancing solidification fronts. The as-cast grain boundaries, therefore, will often consist of a bifilm. Such intergranular features are contrasted with transgranular features as shown in Figure 3. This figure illustrates how the metal is substantially cleaned by the advance of dendrites, but is probably never completely cleaned by this mechanism, and most of the bifilm remnants will have been flattened, stretched out between advancing dendrites.

The preferential precipitation of pores at grain boundaries can now be understood. It is inconceivable that pores could nucleate at a traditional grain boundary because the strain energy required is easily shown to be prohibitively high [9]. It seems clear, therefore, that pores can only form on those grain boundaries which contain a bifilm. Effectively, pores do not form on grain boundaries; they form on bifilms. The ‘air gap’ of the bifilm is easily opened to permit the initiation of a pore.

Similarly, precipitates and second phases will also precipitate on bifilms as favourable substrates. Although, in many circumstances, bifilms will be present both inside the grains and at the grain boundaries, the favoured precipitation at boundaries is often clear. Once again, the precipitation is on bifilms rather than simple boundaries because the strain energy savings from the plastic deformation around the precipitate to accommodate its shape and volume change is much greater than the saving of interfacial energy at normal grain boundaries. We can estimate this benefit as described below.

Consider an inclusion of cubical form, side 10 mm, which is sat on a planar grain boundary. Assuming a difference in surface energies between the boundary and the inclusion of perhaps 0.5 J/m^2^ the energy saving by the precipitate contacting the boundary is simply the area multiplied by the interfacial energy difference = 10 × 10^−6^ × 10 × 10^−6^ × 0.5 = 5 × 10^−11^ J. 

Turning now to assess the condition in which the growth of the inclusion has to force the matrix to yield, assuming for simplicity a yield driven from only the upper face of the inclusion, we have the stress required is the yield stress, perhaps in the region of 500 MPa, so the force required for growth is given by the stress × area = 500 × 10^6^ × 10 × 10^−6^ × 10 × 10^−6^ = 5 × 10^−4^ N. The energy involved is this force times the distance moved = 5 × 10^−4^ × 10^−6^ = 5 × 10^−10^ if the deformation is only 1 mm, but noticing the much wider gaps in bifilms, often around 10 mm, the energy might be as high as 5 × 10^−9^. It follows that the strain energy savings of formation of precipitates are indicated to be 10 or 100 times the energy savings due to economies of interfacial area. 

The favoured location of bifilms at grain boundaries is the result of bifilms being trapped between the random impingement of growing equiaxed grains during freezing. Additionally, in the solid state, the migration of boundaries during recrystallisation and grain growth will be halted at bifilms simply because the migrating boundary cannot cross the ‘air gap’ of the bifilm. Thus, the grain boundary once again becomes coincident with the bifilm. The bifilm *is* the grain boundary.

This contrasts with the interaction of bifilms in the liquid with growing columnar grains. In this case bifilms are often straightened by the ‘pushing’ action of the advancing grains, as illustrated in Figure 3. The result is transgranular bifilms, often lying parallel to lattice planes.

## 6. Eutectic Structure of Al–Si Cast Alloys

This section is based on an article by the author ‘The Two Al–Si Eutectics’, presented at a Symposium held in 2012 in honour of Professor John Hunt [10].

There has been a huge research effort over many years into the structure of the Al–Si eutectic. This has featured John Hunt’s (JDH) central role in the explanation of the structure of the modified Al–Si eutectic as the result of coupled growth of Si and Al, with spacing controlled by physical factors such as interfacial energy and diffusion. As a result, primarily of the work by JDH the structure of the Al–Si coupled eutectic is, therefore, now understood in terms of good physics and in impressive detail.

However, the mechanism of modification, changing the coarse unmodified eutectic into the fine modified form has remained mysterious and an explanation has been elusive. The change of microstructure from a coarse and irregular structure, using an addition of Na or Sr to yield a fine, classical eutectic morphology, is one of those striking, apparently magical metallurgical transformations (Figure 4 and Figure 5) which has fascinated both researchers and the foundry world.

The first valuable clue to a mechanism was the breakthrough publication by JDH and his student at that time, Stephen Flood [11]. By the sectioning of partly frozen samples, it was clear that in the case of the unmodified alloy the formation of the eutectic could occur ahead of the general freezing front, causing the front to appear ragged and somewhat disconnected. This contrasted with the structure after the addition of Na in which the freezing front was now smooth; nucleation ahead of the front had been suppressed. The authors correctly surmised that the unmodified alloy could form on nuclei in the liquid ahead of the freezing front, even though, of course, the nature of the nuclei was not known at that time.

The explanation first put forward in detail by Campbell and Tiryakioglu [12] (the CT theory) based on earlier suggestions [13] and discussions [14] is relatively complex. However, in its defence, and in the opinion of this author, it appears to be the only theory so far that can offer what appears to be a cogent and complete explanation. It explains all the many side effects (some undesirable) of Sr modification and the curious observations made on occasions of apparent modification without the addition of any chemical modifier. For many supporting details and references the reader is referred to the original review [12]. The approach is outlined here, and leads to the clarification of the existence of two quite separate and distinct eutectics in the Al–Si system. The poor relation is the unmodified eutectic (Figure 4), with primary silicon particles displaying coarse, irregular morphology and poor properties, suggested in this account to be the result of impairment by inappropriate industrial processing that generates bifilms. We might, with tongue in cheek, call this ‘The CT Eutectic’. It contrasts with the classical, coupled, strong, eutectic (Figure 5), with fine spacing defined by elegant and admirable physical theory, which might be called ‘The JDH Eutectic’. 

### 6.1. The CT Theory

There seems good evidence that Si in Al–Si alloys nucleates on aluminium phosphide, possibly AlP, or perhaps AlP_3_, as reviewed by Sigworth [15]. We shall not enter into the discussion between the possible phosphides, merely assuming for simplicity in this paper that the nucleus could be AlP.

For hypereutectic Al–Si alloys, with sufficient P addition, it seems that AlP may nucleate freely in the melt, forming a nucleus to which Si can wrap completely around and so develop into a compact crystal; the presence of AlP crystals in the centres of Si crystals, and with clearly related symmetry has been reported repeatedly, and may be detected in Figure 6.

However, in the presence of oxide bifilms, AlP particles nucleate on their outer surfaces. This is assumed since in studies so far, it seems that most if not all intermetallics nucleate on bifilms suspended in the melt [16]. Thus, when Si in turn nucleates on the AlP particle, it is unable to wrap completely around. Instead, it clearly finds the oxide bifilm to be a tolerably favourable substrate, and so will extend outwards, away from the AlP nucleus, spreading across the bifilm (Figure 7). Because of the substantially planar growth morphology of the Si crystal, with its diamond cubic lattice, the bifilm is straightened, changing from its original form as a compact crumpled, fairly harmless morphology resulting from the powerful turbulence during pouring, to become a large flat crack resembling an engineering crack, thereby reducing properties.

The structure of the hypereutectic alloys (Figure 6) is, therefore, a mixture of compact Si particles that have nucleated on AlP particles in suspension in the melt, together with platelet Si that has grown on bifilms. (Interestingly, this dual nature of hypereutectic alloys, consisting of particles and platelets, in which there is competition for nucleation sites in suspension or on bifilms respectively, seems to be generally overlooked in the casting literature). The compact Si particles are formed because the Si is able to wrap completely around the AlP nucleant particle freely in suspension in the liquid. In contrast, the platelet Si is prevented from completely encircling the nucleus because one side of the nucleant is already firmly attached to the bifilm, provoking the Si to grow outwards, spreading across the bifilm (Figure 7). This growth of the Si crystal, simultaneously straightening the bifilm crack, reduces the properties of this eutectic. We call this structure an *unmodified* eutectic.

Sigworth [15] quotes Pechiney work to illustrate that unmodified eutectic is clearly formed at approximately 2.5–6.5 ppm P, whereas at 0.6 ppm P the level is too low to form the unmodified eutectic, indicating that too little AlP is available to act as nucleation sites for Si.

### 6.2. Modification by Sr

The addition of Sr deactivates AlP as a nucleus by forming a layer of Al_2_Si_2_Sr on its surface as discovered by the excellent electron microscopy of Cho and co-workers [17]. Thus, now AlP is rendered inactive, and Si can form neither as primary particles, nor as platelets. The unmodified eutectic at U (Figure 8) can no longer form. The bifilms are now redundant and float about in suspension with no enclosing growth of Si.

In the absence of a favourable nucleation and growth site, the melt now continues to cool until nucleation can occur on some other, clearly less favourable site, possibly on a boride such as CrB as suggested by Felberbaum and Dahle [18]. The lowest temperatures that will favour nucleation and growth will be found adjacent to the walls of the mould. In these regions the new eutectic will now spread, growing continuously as a classical (JDH) eutectic, with its spacing now controlled by diffusion processes. The new eutectic forms at M (Figure 8) having different structure, different composition, different freezing point and different properties. 

The freezing of bifilm-free Al–Si alloys should produce eutectic M without the expense of a Sr addition, and without the disadvantage of porosity. This is an easily tested prediction.

The modified Si has a fibrous, coral-like morphology which is heavily twinned. The twinning has often been supposed to be part of the mechanism causing modification [19]. Here we propose it to be merely a consequence of the individual Si fibres repeatedly being forced to change their growth direction by twinning as a result of interaction with their neighbours. 

This heavily faulted structure contrasts with primary Si that has grown, floating freely on its delicate and pliable raft of double oxide film, and which can, therefore, grow without restraint to achieve a near defect-free structure.

The Al–Si system, having these two quite different eutectics, is a feature of modification phenomena that seems so well known that its significance as two distinct structures seems to have been generally overlooked.

### 6.3. Growth Conditions for the Modified Eutectic

Without good direct evidence, it is a problem to fix the precise composition and temperature of formation of the coupled (modified) eutectic M.

For instance, if we take the traditionally accepted location of U as occurring at 11.5 wt.% Si and 577 °C and accepting the careful temperature measurements [19] of undercooling below U to promote the growth of M being close to 8.5 °C. Assuming a location of M approximately 10 °C under U, and assuming an accurately known and fixed liquidus curve for α-Al formation, we can locate M at the temperature of 567 °C and composition 12.5 wt.% Si. This is midway in the range of scatter of eutectic temperatures and composition seen in currently quoted Al–Si phase diagrams.

A totally different approach to estimating the location of M might be to assume that a peak in fluidity would be expected to be observed at the coupled eutectic as indicated by much evidence in the literature on the fluidity of binary alloys; in all studies of binary alloy systems the fluidity peak and the eutectic composition accurately coincide. For Al–Si alloys, most the data confirm [16] that a fluidity maximum occurs at somewhere between 14 and 18 wt.% Si as illustrated from one set of data taken as an example in Figure 9. If the fluidity peak truly indicates the eutectic location in the Al–Si system, by cross-checking with Figure 8, the temperature of M would be between approximately 550 and 510 °C. These extremely low temperatures seem unlikely. However, if the fluidity peak and eutectic do not coincide in the Al–Si system, this would correspond to exceptional alloy behaviour and would not be easily explained. More careful research is needed to clarify the eutectics U and M.

### 6.4. Porosity Development by Sr

After the addition of Sr, the bifilms are no longer trapped inside Si platelets because they are no longer employed as substrates for the growth of Si particles. Thus, the bifilms remain freely floating, suspended in the liquid. They are now free to wreak damage in different ways. For instance, they will now be expected to be drawn down into the dendrite mesh, or into the regions between grains of eutectic, where they can block the flow of interdendritic liquid and so limit feeding and instead grow porosity. In a simple and elegant experimental study, Fuoco [21] clearly confirms the rather shocking fact that an Al–7Si–0.4Mg alloy has some modest permeability, but when modified by Sr the permeability drops to zero; there is no interdendritic flow.

Knowing the nature and presence of bifilms allows us to predict that the modified eutectic will be in danger of the development of porosity. Not only is interdendritic feeding no longer possible because of the blockage of interdendritic flow paths, but the presence of bifilms means that their doubled nature will allow them to open under the consequential reduction in pressure on their ‘downstream’ side, their downstream film being pulled away to open up porosity. 

It is disquieting to realise that the formation of porosity will be perhaps the least of the problems for the Sr-modified eutectic, since the main problem will probably simply be the presence of a huge population of bifilms, acting as cracks. The bifilm population will become clear if the hydrogen content or shrinkage conditions are sufficient to unfurl and expand them into pores. Unfortunately, the growth of pores will be aided by any increase in hydrogen content, and increased hydrogen is probable; it will be encouraged by the enhanced reactivity of the Sr-containing melt with its environment [12].

Thus, the Sr-modified eutectic, although in itself intrinsically strong and tough, will have its strength properties significantly reduced by the release of the bifilm population, and possibly further reduced by the bifilms opening to become pores. The properties of the modified eutectic will, therefore, be significantly controlled by the prior quality of the melt. This prior quality in terms of oxide population is usually never controlled adequately, with the result that Sr modification is sometimes seen to improve properties, but in other foundries is seen to impair properties. A number of Al alloy foundries have, therefore, abandoned modification by Sr as offering a beautifully fine microstructure, but at best no improvement, and more often, a reduction of their strength properties [12].

### 6.5. The Coupled Eutectic Zone

The author has a concern that there has been some doubtful science and muddled thinking about the existence of the so-called asymmetric ‘coupled zones’ for the successful growth of the modified eutectic, which, frankly, he has never been able to understand (the author admits he may be at fault here!). However, the presence of separate zones under each eutectic, under the crossed liquidus lines (Figure 8), is perfectly straightforward and reasonable, and illustrates the probability that the existence of the so-called asymmetric or skewed coupled zone is in fact a natural confusion arising from the presence of two overlapping undercooled zones.

### 6.6. Alternative Modification Processes

In addition to Sr, it is well known that there are a number of other chemical elements such as Na, Ca, and Mg. that can cause modification of the Si phase in Al–Si alloys. It seems likely that a chemical mechanism similar to that of Sr will also apply.

However, there are also many references in the literature to modification by physical and mechanical processes not involving the additions of chemical elements. These include ultrasonic, electromagnetic stirring and mechanical vibration [22]. Jorstad [23] has reported the automatic modification of eutectic liquid in partly liquid alloys. Once again, these reports seem initially surprising and mysterious, but they appear to be explicable assuming a bifilm hypothesis. The eutectic observed between grains of solid in so-called ‘semi-solid’ casting appears to be modified, despite the absence of modifiers such as Sr or Na. This behaviour is reported by Jorstad [23] only for mixtures that contain over approximately 50% solid. For such high proportions of solid, conditions in the liquid between solid particles would be expected to be brutally turbulent: the bifilms being constantly impacted by solid material, or violently squeezed out as solid grains move together and liquid and its suspended bifilms in the gap is expelled. Modification, therefore, is surmised to be a result of the inability of bifilms to unfurl in conditions in which they are denied the tranquil conditions they require to unfurl; they are unable to straighten to provide the flattened substrate on which primary platelets of silicon can grow. 

Modification by chilling is commonly reported. Although this might be explained simply by the common observation that all cast microstructures are refined by faster cooling, this curt attempt to dismiss the problem cannot explain the distinct change in structure from the irregular eutectic to the classical eutectic form. Assuming the presence of bifilms, and the necessary assumption of a rate dependence of unfurling against the viscous restraint of the liquid, the explanation seems likely to be that the bifilms often take minutes to unfurl and will now no longer have time to open [2]. Thus, primary Si particles no longer have an easily flattenable substrate on which to extend, and so will be discouraged from growing, in favour of growth as a classical regular eutectic at lower temperature. The structure produced by the new Ablation Casting Process (Figure 2) is a good example [24]. (Ablation is a new process in which a sand mould bonded with a soluble binder is dissolved away by water jets, driving extremely rapid freezing of the casting by direct impingement cooling—avoiding the normal resistance to heat flow posed by the air gap between the casting and mould).

Interestingly, Gruzleski and Closset [19] deny that quench modification is the same as impurity modification by Na or Sr, even though these two approaches to modification result in the same fibrous morphology of the Si. They quote work on the defect structure of the eutectic Si in quench modification drawing attention to the low level of twinning defects, rather similar to the level of defects in unmodified eutectic Si. A low level of defects in chilled material might be expected as a result of the strong temperature gradient, aligning growth of the Si fibres, so that little twinning is necessary to repeatably re-align fibres as they grow, they no longer have to ricochet so often between their neighbours.

In an Al–Si hypoeutectic alloy that contains no bifilms, growth of primary Si platelets should not be possible. Thus, the alloy should automatically solidify as a modified eutectic.

This prediction has been confirmed by Chinese workers [25,26] using an Al–Si alloy that had been obtained directly from a mixed electrolysis of Chinese bauxite. They found they could retain the modified structure up to an amazing 16 wt.% Si. If they had a clean alloy with higher Si they could probably have retained the modified structure to much higher Si levels. However, in their work the modified structure appeared to be lost at 17 wt.% Si. This was almost certainly because to progress beyond 16 wt.% Si they added a master alloy containing 50%Si (without doubt containing generous quantities of oxide bifilms to facilitate the growth of primary Si plates).

Jian et al. [27] found they were able to produce the modified structure by the application of ultrasonic vibrations to the melt. At this time, it is unclear whether the ultrasonic treatment was effective in eliminating bifilms or, more probably, the rapid stirring action by the ultrasonic jet preventing unfurling. Similar results have been observed by rapid rotation of the melt [28].

### 6.7. Future Potential for Al–Si Alloys

Melts devoid of bifilms seem to be attainable without difficulty or expense, but, of course, reports of confirmatory experimental work would be welcome. Although, as noted above, they might be produced by ultrasonic treatment or centrifuging, these are not likely to be useful future processes.

Suppliers of primary metal should be capable of supplying ingots of substantially clean Al and Al alloys; the potential for the profoundly different behaviour of clean alloys is illustrated by Wang [26]. However, it seems at this time producers seem unaware of any advantage to providing such supplies.

A further option relating to metal supplies might be the possibility of obtaining melts with P levels below 0.5 ppm, so that even though bifilms are not removed, they have no resident nuclei to initiate the formation of primary plates, forcing the Si to precipitate at a lower temperature in a modified form (eutectic M in Figure 8) [15].

Alternatively, accepting our current supplies of dirty metal and alloys from existing suppliers, the treatment of the melt with Ti-B grain refiners is a known method to encourage heavy titanium borides to precipitate on bifilms and sink them to the bottom of the melt. If undisturbed, the clean melt can be decanted off, leaving the sediment in place [2].

Foundries would benefit from the development of techniques for the reduction of the oxide bifilms in their melts. In proportion to the reduction in bifilms, Sr addition can be reduced. If bifilms can be reduced to zero or near-zero (in the view of the author a perfectly practical proposition) Sr can similarly be reduced to zero. However, the consequential additional benefits to clean metal will be of far greater importance than the cost saving of Sr.

## 7. Textures

The development of textures in worked metals was surveyed by Hutchinson in 2015 [29] who presents an interesting and wide-ranging review of the possible causes of anisotropy in metals. However, interestingly, and perhaps predictably, he concludes that a better understanding would be a suitable area for further investigation. It is significant that his review did not include the presence of bifilms which appear to be present in most of our engineering metals.The inclusion of bifilms would almost certainly have provided that final piece of the puzzle that he lacked.

As we have already discussed, any migrating or recrystallising boundaries which encounter a bifilm will be pinned in place. In a sense, there is an obviousness about this effect, as typified by the numerous experiments to bond sheets of Al alloys. As a clear instance, foil laminates of 7475 aluminium alloy were stacked and press bonded to a reduction of between 10:1 and 15:1. The consolidated laminates are observed to experience grain growth inhibition by the presence of the two oxides on each of the bonding surfaces, acting as a robust bifilm crack [30]. A less robust oxide film on copper sheet, repeatedly folded and rolled to give 256 layers subjected to a total strain of 4.8, followed by recrystallization, showed a similar strongly influenced texture, illustrating that the control of textures by oxide bifilms is likely to be widespread [31].

Exfoliation corrosion appears to be another bifilm phenomenon, being a form of intergranular corrosion associated with Al alloys which have been rolled or extruded to produce an elongated grain texture [32]. The corrodent can ingress those bifilms which happen to intersect and emerge at the surface of the metal, penetrating and spreading along the unbonded central interfaces of the bifilms and corroding away their residual points of attachment, until whole flakes of alloy detach from the surface. 

It seems likely, therefore, to this writer that the near-universal presence of bifilms currently in our metals is one of the most important, if not the most important, contributor towards the development of anisotropy and texture in metals. Interestingly, because it is now possible to produce metals free, or nearly free, from bifilms, critical tests of their effects should, in principle, be easy to apply.

## 8. Elongation

Tensile elongation is especially sensitive to the presence of prior cracks (Figure 10). The effect is clearly of major importance [33]. Although the foundries and laboratories which determined these data in Figure 10 would claim their casting methods were under strict control, the immense variability of results indicates a serious problem not controlled by traditional foundries. Most probably the uncontrolled aspect was the bifilm population—a result of poor metal quality, further degraded by turbulence during pouring of the casting. There is now solid shop floor foundry experience of the truth of this assertion. It would be useful for scientists to catch up with this important finding and quantify it with careful laboratory research. In Figure 10, at a yield strength of approximately 300 MPa the elongation ranges between approximately 3 and 18 per cent. These historical results contrast with reproducibly high elongations now routinely achievable when using good casting technology designed to reduce the entrainment of air and oxides during mould filling. 

These results for sand moulds, in which freezing rates are rather slow, are much lower than those achievable for permanent moulds. The faster freezing traps the bifilms in their compact, convoluted shape immediately after the turbulence of the filling of the mould, and so maximising tensile properties for that population of bifilms.

The upper boundary shown in Figure 10 was originally thought to be a maximum. As a result, it was for a time used as a quality index, proposing the boundary to be 100% quality, and an elongation result of only half of this value (giving a quality index of 50%) being a good result for many foundries. 

Later work, however, discovered sources of wrought Al alloys [34] which had elongation/yield properties which significantly exceeded the boundary, causing its fundamental nature to be questioned. It may be that as bifilm populations become very low, sand cast and investment cast Al alloys may also exceed the boundary. 

Even so, the appearance of the boundary, perhaps indicating practical reasonable limits to the property regime for cast Al alloys, is interesting; it suggests higher strength alloys may become practical because experiments to develop improved alloys easily achieve high strength but the alloy is a failure because it has become too brittle. The height of the boundary allows us to speculate that even if strength were somehow greatly increased, provided the bifilm content were low, toughness should be remain more than adequate. For instance, extrapolation of the boundary indicates a potential yield of 400 MPa but retaining 12% elongation, or even 500 MPa with approximately 8% elongation. There seems to be valuable potential for future strong and tough Al–Si cast alloys. 

## 9. Fatigue

The classical fatigue failure, in which the striations, indicating the successive positions of the fatigue crack, start at a tiny defect, but spread over most of the fracture surface, finally fracturing by overload, is not so common. More common is the fracture surface for which a few grains appear to show some striations, but striations are absent for the majority of the fracture surface.

For castings which have solidified with significant area of columnar grains, these commonly fracture as facets because of the presence of transgranular bifilms flattened by the mechanism shown in Figure 3. Even so, the facets shown in Figure 11 are so perfect that it is difficult to be sure that they are flattened bifilms; they may originate by a slip-plane mechanism as has been traditionally assumed. 

However, the facet in Figure 12 can certainly be identified as formed by a flattened oxide bifilm, because it is continuous with the clear image of an oxide film (even clearer in a magnified image) wrapped around the sand inclusion. Furthermore, its microscopic undulations and imperfections are probably to be expected of a bifilm flattened by the slow and imperfect progress of advancing columnar grains during freezing.

In both Figure 11 and Figure 12, the non-faceted area, the fracture surface surrounding the facets, are almost certainly formed by biofilms—the detail of the structure is not convincingly metallic, but is more convincingly that of a highly wrinkled and convoluted mass of bifilms. Thus, on this fracture surface there are planar features and rough features, but all features are probably bifilms. There probably is no metal to be seen! Further careful study of such surfaces is required to provide some certainty to these outstanding and important questions. The use of ‘pull-pull’ type fatigue loading (positive R ratio) is strongly recommended for such research; the crack remains open and the fracture surface undamaged, preserving its original details, rather than the fracture surfaces being repeatedly closed so as to damage the surfaces by impact and shear. Furthermore, of course, the pull-pull mode of fatigue is a better simulation of most real fatigue conditions in which a residual tensile stress in the component aids the initiation and progress of fatigue.

Özdeş and Tiryakioglu have investigated the relation between elongation and fatigue in cast 319 alloy [36]. They find a powerful link as typified by Figure 13. The work reveals that fatigue life need not be tested by a lengthy fatigue test but can reliably be estimated from a simple and quick measure of elongation from a tensile test. This is a hugely important and valuable finding for this complex Al–Si alloy which should be verified for a wider variety of Al alloys, especially solid-solution alloys, and perhaps other metals such as irons and steels. 

The other important message of Figure 13 is that because bifilms are known to control elongation, elongation in turn seems to control fatigue and, therefore, bifilms control fatigue. 

With regard to the interpretation of the fracture surfaces of fatigued Al–Si alloys, Murakami [37] notes that cracks appear to initiate from (i) cracked primary Si particles or (ii) decohesion of the Si particle interface with the matrix or (iii) so-called shear failures of the matrix. All these initiation modes are consistent with the participation of bifilms: (i) the central crack in the Si particle is the bifilm around which the particle formed; (ii) if the particle formed on only one side of the bifilm then the opening of the bifilm would appear as a decohesion from the matrix; and (iii) failures in the matrix are to be expected if a proportion of bifilms are present as free-floating defects on which Si has not precipitated. 

It is curious how bifilms seem so central to mechanical properties but have been unknown for so long. 

## 10. Hydrogen Cracking and Blistering

An environment of dry hydrogen seems inert to Al and its alloys, but contact with water or water vapour leads to dissolution of hydrogen in the metal [6]. This is well known to heat treaters of forgings because blisters can form during heat treatment if the atmosphere is moist, which it often is, as a result of the basket containing the heat treated parts having been used at some previous time for quenching parts into water. Preventing the uptake of hydrogen by provision of fluorides which create fluoride vapours in the furnace is effective [6] but not popular as a result of the toxicity of fluorine, although low levels of sulphur dioxide gas seem also to be effective [38].

Forgings seem more sensitive to blister formation than castings because the mechanical working of the metal tends to straighten the bifilms, so that those sufficiently near to the surface become more parallel, and gain area, so more easily expand as their internal pressure of hydrogen increases, and the metal softens with the raised temperature (Figure 14).

The probable mechanism of hydrogen embrittlement, in which hydrogen uptake results in serious loss of mechanical properties, has been outlined elsewhere [40]. It is described briefly here. 

It seems likely that hydrogen enters the metal faster than its 10^−8^ m^2^/s rate of diffusion would suggest, by penetrating along the ‘air gap’ of bifilms as easy pathways into the interior. Those bifilms not linked to the surface will fill with hydrogen by diffusion from the nearest pathway. The pressure in the bifilms will rise to equilibrate with that pressure generated by dissociation of hydrogen gas into hydrogen atoms. If, as is normal, the bifilm population is dense, the increasing internal pressure, acting over the total area of bifilms, will act to counter strength and ductility, so both strength and ductility will suffer, and may both be reduced to zero.

This behaviour can be modified by heat treatment. The strain energy arising from the shape and volume change accompanying the arrival of a new phase can be greatly reduced if the new phase forms on a bifilm, because these elastic and plastic strains can be more easily accommodated by the growth of the precipitate into the air gap, as has been discussed earlier. This energy saving is up to 100 times greater than the interfacial energy savings of forming the precipitate on a grain boundary. Thus, precipitates will be expected to form not on grain boundaries, but only on bifilms. (The fact that a high proportion of bifilms will be at grain boundaries explains the common impression that precipitates favour grain boundaries). The bifilm is forced open somewhat by the trespassing of a percentage of the expanding volume of the precipitates into the air gap. The increased internal volume provided by the expanded air gaps acts as major sinks for hydrogen, initially lowering the concentration of diffusing hydrogen in the metal. However, eventually, the pressure in the bifilms may reach the yield strength of the metal, at which point the metal will yield under negligible externally applied stress. 

Blisters tend to form during hydrogen charging experiments to study hydrogen embrittlement, and the researchers have lamented how this has confused or undermined their experiments because it has not been clear how blisters could nucleate and grow in their experimental conditions [41].

If we have successfully highlighted the mechanism of hydrogen embrittlement, it follows that the phenomenon is easily solved by eliminating bifilms. This is surely another prediction which could be easily and quickly confirmed.

## 11. Conclusions

Bifilms acting as cracks appear to be prolific in aluminium alloys, and significantly control aspects of both microstructure and properties. 

The presence of bifilms can be controlled using newly developed techniques for the handling and casting of liquid metal [2]. However, it has not been possible to include these techniques in this short review. This could be the subject of a further review in due course. 

By implication and extrapolation of the research findings described in this report, good process control of bifilms appears to have the potential to revolutionize our interpretation of microstructures and the potential to revolutionize the properties of aluminium alloys. Confirmation arises from foundries who now practice good bifilm control, and are routinely achieving good, previously unattainable, properties as a direct result.

## Figures and Tables

**Figure 1 materials-14-01297-f001:**
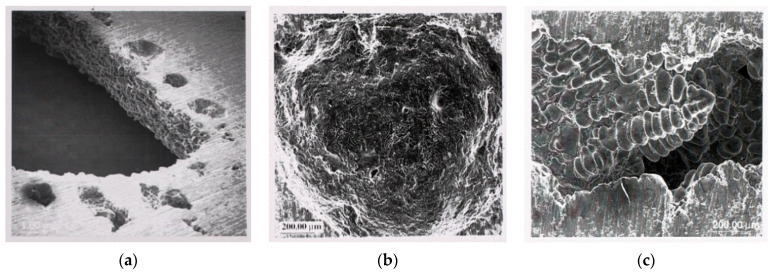
Hydrogen gas diffusing away from a core (**a**) precipitates early in (**b**), but delayed in (**c**) because of more difficult inflation of bifilm (Courtesy S. Fox 2000).

**Figure 2 materials-14-01297-f002:**
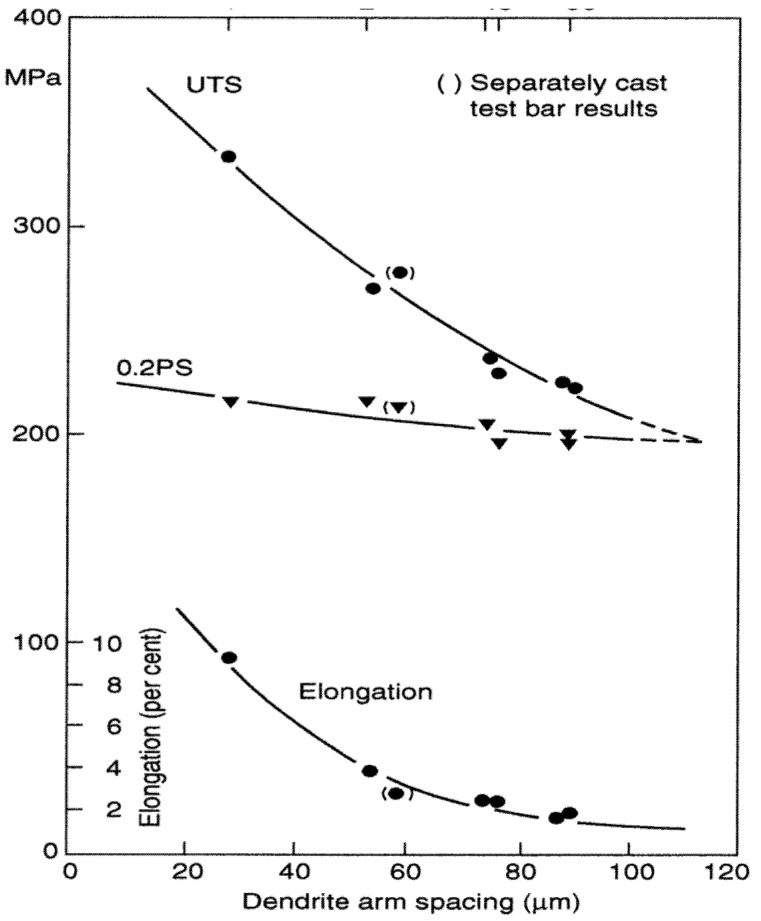
The effect of DAS on tensile properties for Al–7Si–0.4Mg alloy [7].

**Figure 3 materials-14-01297-f003:**
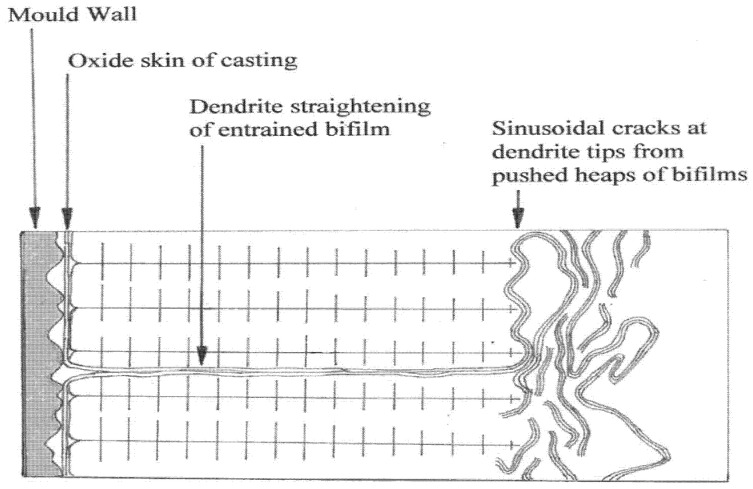
Dendrite ‘pushing’ of bifilms to created transgranular cracks and facets [2].

**Figure 4 materials-14-01297-f004:**
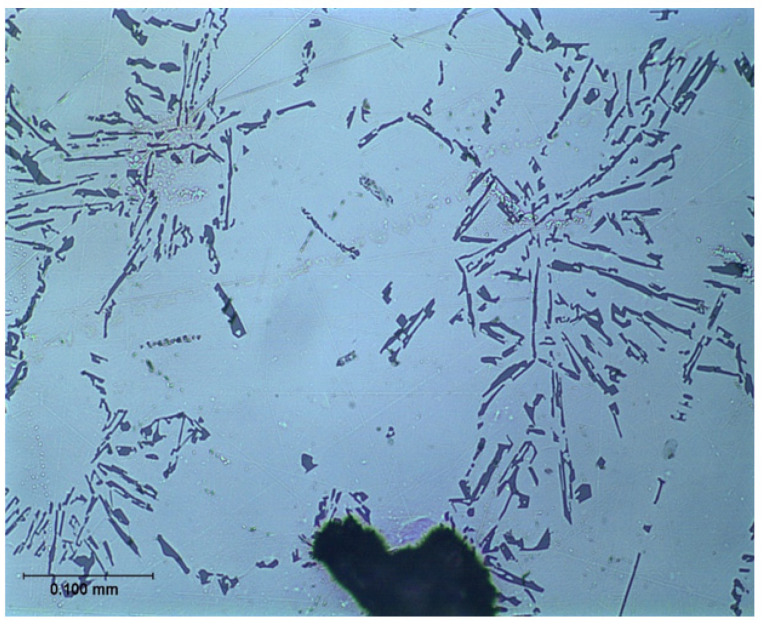
Unmodified Al–7wt.%Si alloy (courtesy Alotech Inc., Cleveland, OH, USA).

**Figure 5 materials-14-01297-f005:**
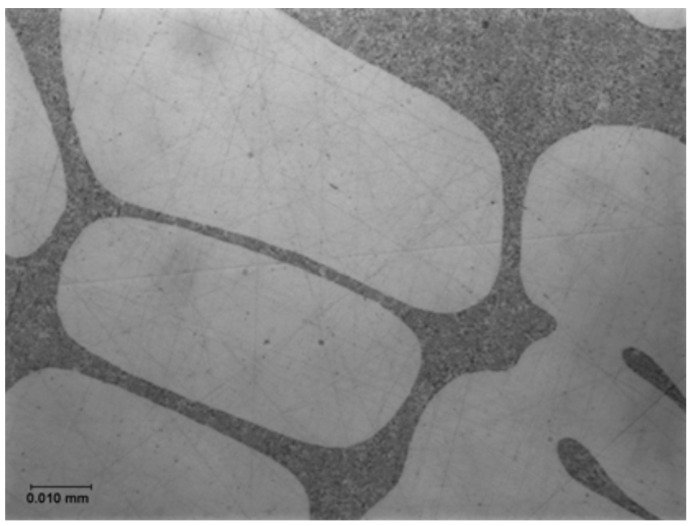
Ablation-modified Al–7wt.% Si with interdendritic classical coupled eutectic. Too fine to be resolved at 1000× magnification (courtesy Alotech Inc., Cleveland, OH, USA).

**Figure 6 materials-14-01297-f006:**
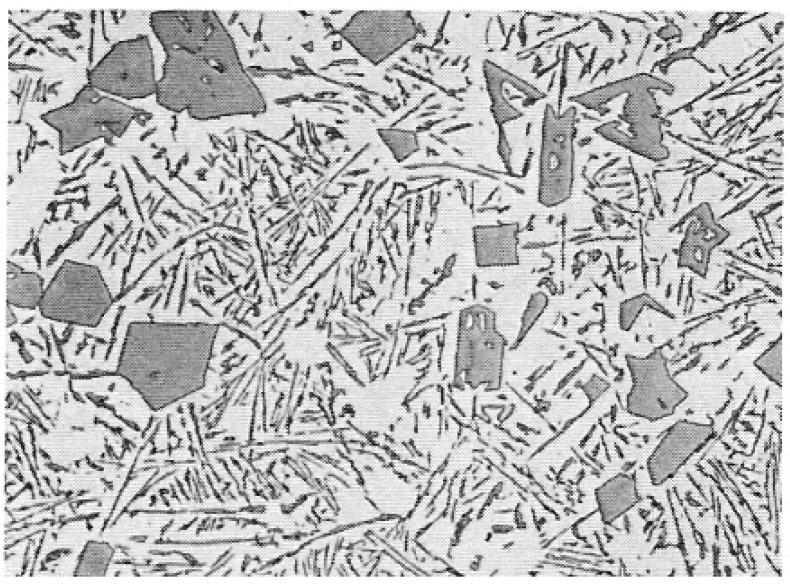
Hypereutectic alloy showing primary Si cyrstals and unmodified Si eutectic matrix.

**Figure 7 materials-14-01297-f007:**
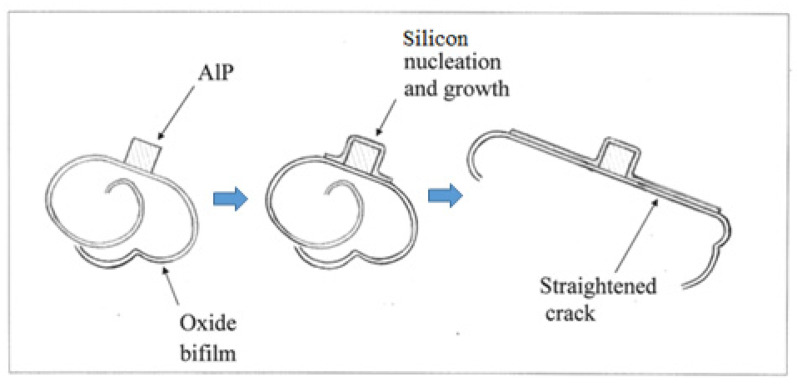
Nucleation of an AlP particle on a compact bifilm,.which in turn nucleates Si, whose planar growth straightens the bifilm.

**Figure 8 materials-14-01297-f008:**
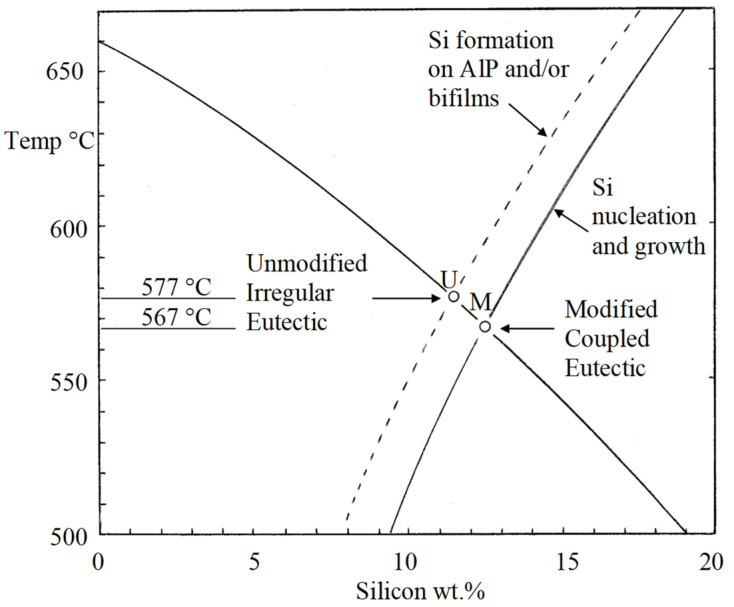
Al–Si phase diagram illustrating the two separate eutectics.

**Figure 9 materials-14-01297-f009:**
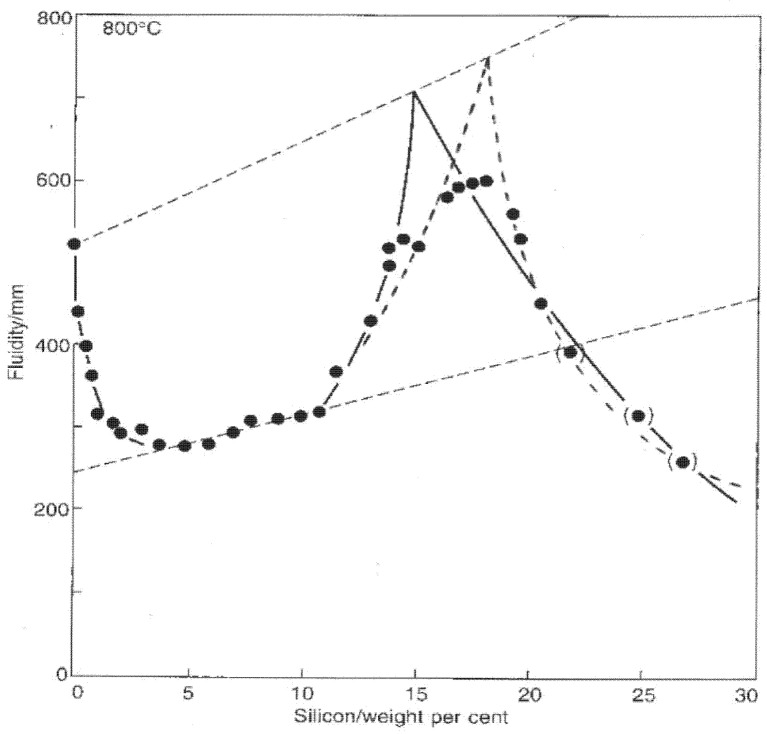
Typical fluidity results for Al–Si alloys showing a peak somewhere between 14 and 18 wt.% Si (data from Lang [20]) (the suggested linear limits are by the author [2]).

**Figure 10 materials-14-01297-f010:**
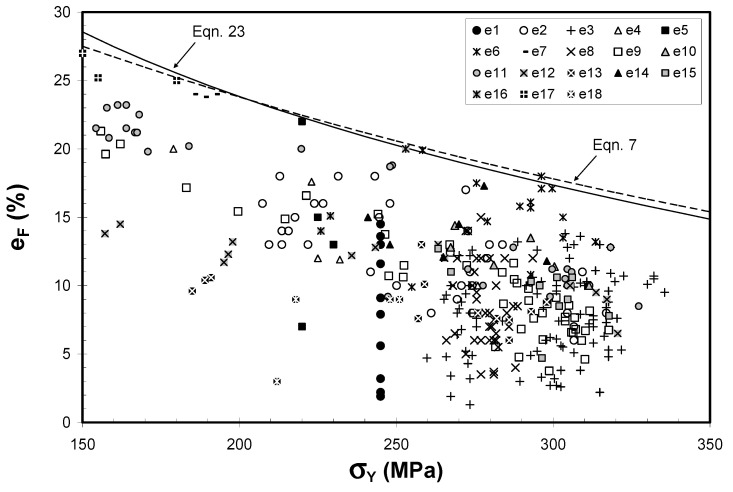
A survey of 18 published results for tensile elongation and yield strength of cast and heat treated Al–7Si–0.4Mg alloys from mainly aerospace foundries [33].

**Figure 11 materials-14-01297-f011:**
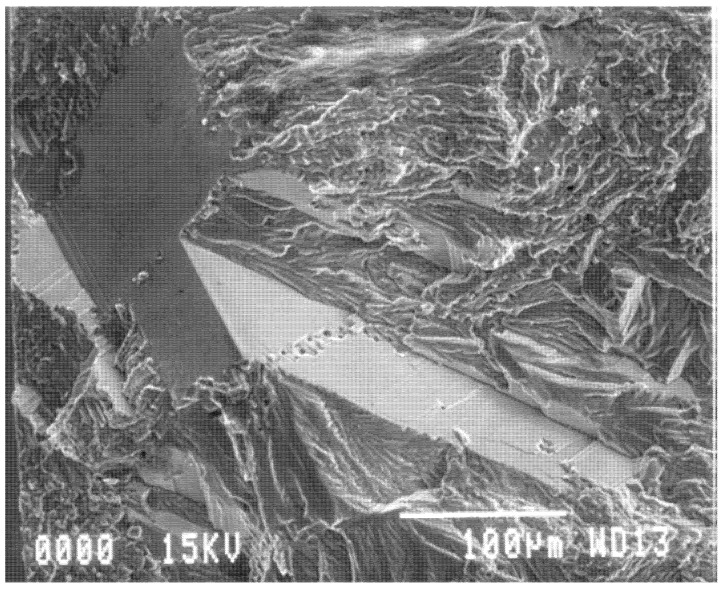
Fatigue fracture of Al–7Si–0.4Mg alloy showing especially perfect facets [35].

**Figure 12 materials-14-01297-f012:**
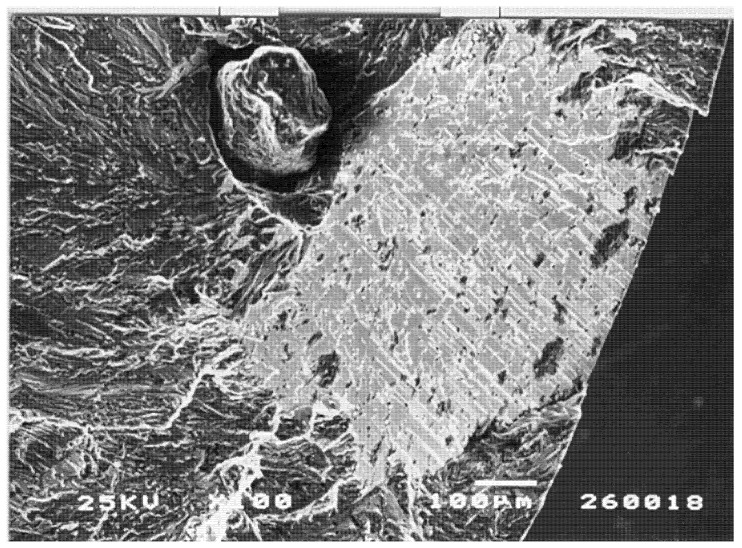
Fatigue fracture of Al–7Si–0.4Mg alloy showing facet generated from the oxide bifilm partly wrapped around the entrained sand grain [35].

**Figure 13 materials-14-01297-f013:**
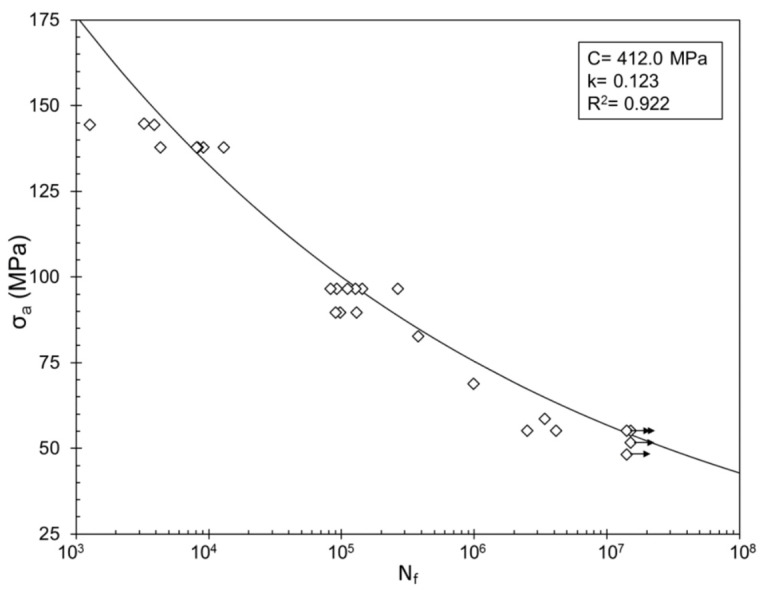
Relation between quality factor (mainly based on elongation) and fatigue life of cast 319 alloy [36].

**Figure 14 materials-14-01297-f014:**
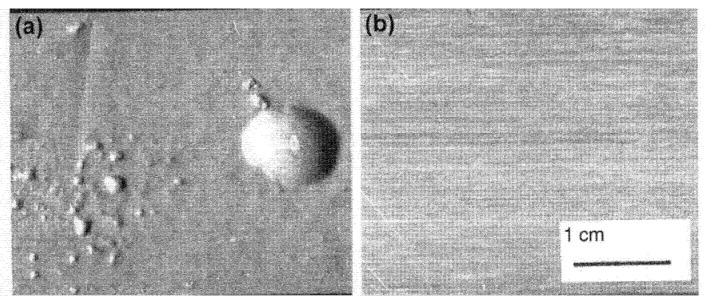
Surfaces of stretched sheet of 7475 (Al–Zn-Mg-Cu) alloy heated to 515 °C (**a**) in air; (**b**) in air containing some NaBF4 [39].

## Data Availability

Data sharing is not applicable to this article.

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
