# Peer review of "A Personal View of Microstructure and Properties of Al Alloys"

_materials, 2021, doi:10.3390/ma14051297_

Round 1

Reviewer 1 Report

Dear Author,

Congratualtions on your work, which represents a personal view about an interesting issue. However, the paper presents a mix structure between a paper review and a research review, leaving the reader a little bit confused.

Thus, I'm suggesting the following actions:

At the end of the Introduction, please include the way as the paper is organized, letting the reader knows what can be found in each section.

After the Introduction, please include a METHODOLOGY section describing how did you developed this work, allowing other to make the same about other issues.

Please consider to revise the sections distribution. Some sections are too short and must be developed or cut.

The Conclusions are disappointing. After a goor work like this, some bullet points are expected by the reader, alloying to understand the main achievements brought by the development of this work, mainly in a structured manner. Thus, please consider to improve this part of your paper.

The sentence: "Good process control of bifilms appears to have the
potential to revolutionize our interpretation of microstructures and the potential to revolutionize the properties of aluminium alloys." should be properly supported by the development of the manuscript.

Kind regards,

Reviewer 

The importance of bifilms should be highlighted in the title of the paper.

Author Response

The interesting and courteous critique by the reviewer is appreciated, even though some of the proposed requirements set the author significant challenge.

At the end of the Introduction, the suggestion to precis the contents of the sections is a good idea, and I have endeavoured to provide this.
The presentation of a methodology is quite another matter, however. I have no idea how to do this. I had no method. The paper came straight out of my head and straight on to the page. This is how I write. I would request the reviewer's patience and understanding on this issue, because I took on the task of writing this overview for mdpi on the understanding that the article was to be a personal view.

With regard to her/his point on length of sections, I very much trust the reviewer will agree that the importance of each section lies in whether they have anything to say, rather than whether they are short or long. I would therefore wish to defend the continued acceptance of the short sections. Even so, I have made a small but important addition to the short section 4.

I agree with the reviewer on the shortcomings of the Conclusions, and have sought to rewrite them. I hope the reviewer will find them somewhat improved. I have found any further improvement elusive because of the breadth of the subject matter. A final sentence to draw attention of the importance of the subject to world-wide application is not, I think, out of place, even though I know the reviewer may fault me on the convention that Conclusions usually relate only to the subject matter discussed in the paper. I hope the reviewer will regard this as a very minor transgression.

With respect

John Campbell

Reviewer 2 Report

In this manuscript which presents a personal view by the author of the role of biofilms in Al alloys, porosity, microstructure and mechanical properties of this class of alloy have been discussed. The manuscript is very well organized and written and thus it is recommended for publication. However, before the publication the following issues should be considered and addressed:

  1. There are some Figures such as Figure 1, Figure 3, etc. that need a reference as well as copyright permission for reuse those.
  2. There are a lot of references that can not be found or they have not been well cited that should be replaced. However, in general, the references in this manuscript are not according to the template of Materials. Besides, instead of very old works, it is recommended to include some recent works that have been worked on the advanced characterization methods.

Author Response

Thank you for your kind words about the submission, and your recommendation that it should be published. Your concerns are addressed in order.

Figures 1 and 3 have been provided with references. 

I have already requested help from the editorial department to secure permissions.

I shall endeavour to put both the ms and the references into the approved house style.  However, the concern which the reviewer makes regarding the age of the references is not easily addressed, simply because they reflect rather accurately the age of the author. I do believe, however, that despite their age, the references generally remain important, being significant milestones in their time (and many not yet appreciated to this present day).  Further, it is a sobering thought that the age of the research which is referred to here has also be heavily influenced by the simple fact that research on casting was carried out years ago, but has more recently become unfashionable.  As the reviewer’s comments imply, modern casting research is correspondingly thin on the ground, to the detriment of metallurgy as a whole.

Reviewer 3 Report

Submitted manuscript entitled “A personal view of microstructure and properties of Al alloys " described in details a personal view by the author of the role of bifilms in Al alloys. Topic is very interesting and I recommend it to publication. I think paper is written in good language. Technique, technology and research methods used in the work are adequate. Methods and obtained results prove founded thesis and show originality of the manuscript.

Minor remarks

Some of the figures (microstructures, graphs): figure 1; figure 2 has already been published in others publication of the author: An Overview of the Effects of Bifilms on the Structure and Properties of Cast Alloys; DOI: 10.1007/BF02735006. and MATERIALS PERSPECTIVE Entrainment defects; DOI: 10.1179/174328406X74248

Page 2, line 81: Figure 1, (a), (b) and (c) markings in the figures is missing

Page 3, line 98: Figure 2, very poor quality of the figure, can it be redrawn

Page 17, line 586-687: Figure 14, different font size in drawing caption

Author Response

The author appreciated to receive the kind introductory words by the reviewer. With regard to the concerns raised, the points are dealt with successively.

It is true that a number of images used as figures in the text have been used by me before.  However, they are inserted for the purpose of illustrating the text, and, I think, the text would be poorer without them.  The ms is not intended as a piece of original research, but merely a piece of original thinking around a wide subject in the light of new logical constructs (bifilms) which have been identified and which appear widely important with respect to their effects on the microstructure and properties of Al alloys. 

The individual images have been identified in Figure 1, thank you.

With respect to Figure 2, I regret to have to admit that my drafting skills are very limited, and I do not think I am able to offer any improvement.  However, importantly, I do think it is clear (if awful).

The font of the caption to Figure 14 is corrected.

Round 2

Reviewer 1 Report

Dear John Campbell,

Thank you so much for your sincere answer to my questions.

Indeed, with your age and knowledge, you deserve to publish what you want.

I would like to do the same with your age and brilliant career.

I wish you a healthy and happy year of 2021.

Kind regards.